# Guard Cell Membrane Anion Transport Systems and Their Regulatory Components: An Elaborate Mechanism Controlling Stress-Induced Stomatal Closure

**DOI:** 10.3390/plants8010009

**Published:** 2019-01-03

**Authors:** Shunya Saito, Nobuyuki Uozumi

**Affiliations:** Department of Biomolecular Engineering, Graduate School of Engineering, Tohoku University, Aobayama 6-6-07, Sendai 980-8579, Japan; s.saito@biophy.che.tohoku.ac.jp

**Keywords:** guard cell, drought stress, salt stress, bacterial immunity, anion channel, protein kinases, calcium signaling, abscisic acid signaling, ion homeostasis

## Abstract

When plants are exposed to drastic environmental changes such as drought, salt or bacterial invasion, rapid stomatal movement confers tolerance to these stresses. This process involves a variety of guard cell expressed ion channels and their complex regulation network. Inward K^+^ channels mainly function in stomatal opening. On the other hand, guard cell anion channels play a crucial role in the closing of stomata, which is vital in terms of preventing water loss and bacterial entrance. Massive progress has been made on the research of these anion channels in the last decade. In this review, we focus on the function and regulation of *Arabidopsis* guard cell anion channels. Starting from SLAC1, a main contributor of stomatal closure, members of SLAHs (SLAC1 homologues), AtNRTs (Nitrate transporters), AtALMTs (Aluminum-activated malate transporters), ABC transporters, AtCLCs (Chloride channels), DTXs (Detoxification efflux carriers), SULTRs (Sulfate transporters), and their regulator components are reviewed. These membrane transport systems are the keys to maintaining cellular ion homeostasis against fluctuating external circumstances.

## 1. Introduction

When exposed to saline or water-deprived condition, plants respond in various ways to increase their survival rate. Stomatal movement is one of the key features of this response. Whenever plant senses salt or drought stress, and the consequent demand for saving water, rapid stomatal closure is induced to prevent water loss. In addition, pathogen attack can also trigger stomatal closure, for stomata can be a pathway for bacterial infestation. It is well known that light induced stomatal opening is controlled by inward-rectifying K^+^ channels localized in guard cell PM (plasma membrane), such as KAT1, KAT2, AKT1, and AKT2 [1,2,3,4,5,6,7,8]. In contrast, stomatal closure consists of guard cell expressed outward K^+^ channels and anion channels. Particularly, when plants are exposed to the stressful conditions mentioned above, activation of guard cell anion channels holds the key to defending themselves by inducing rapid stomatal closure.

Plant cells contain several types of anions including chloride, nitrate, sulfate, and organic acids like malate. Chloride is a major component of salt in soil, alongside with sodium, a major inflictor of salt damage to plants. While chloride acts as an essential nutrient, accumulation of chloride in the shoot (even without the presence of sodium) causes a decreased rate of transpiration and photosynthesis, leading to reduced crop yield and quality [9,10,11]. Nitrate works as an essential nitrogen source for amino acid synthesis. The process starting from direct uptake of nitrate from the soil, followed by enzymatic reactions and consequent production of glutamate, is an exclusive feature for plants (nitrogen assimilation) [12,13]. Nitrate also acts as an antagonist against chloride and could be applied to prevent Cl over-accumulation in shoots. Malate is important as an intermediate of TCA-cycle, essential storage carbon molecules and major photosynthate in CAM and C4 plants [14,15,16,17,18,19,20,21,22,23]. It also participates in the biosynthesis of amino acids and fatty acids, root growth, and aluminum tolerance [24,25,26]. Sulfate is an essential source for the biosynthesis of cysteine. Cysteine can either be directly incorporated into a protein or a peptide such as glutathione (GSH), or can be used as a sulfur donor for various coenzymes like molybdenum cofactor required for ABA synthesis [27,28,29,30].

These anions, once produced or uptaken, are immediately transferred to appropriate tissues and cell compartments by various anion transport systems, anion channels and transporters. The anion membrane transport system is distributed in each cell of the entire plant, and, in particular, many types of anion channels function intensively in guard cells, since the cell volume fluctuates in response to frequent environmental changes. The mechanism of stress-induced stomatal closure thus relies on these guard cell expressed anion channels. Under stress conditions, they drive anions either outside the cell or into the vacuole, triggering change in cell turgor pressure and consequently reducing the volume of guard cells. In this review, we focus on the function and regulation of such anion channels in guard cell.

## 2. SLAC1, a Major Contributor of Stomatal Closure

Early patch clamp studies in the 1980s revealed two types of anion channels present in guard cell PM: R (rapid)-type and S (slow)-type. R-type channel activated rapidly within 50 ms by depolarization, while S-type channel showed slow voltage-dependent activation and deactivation [31,32,33,34]. Then, 2008 saw a breakthrough: an ozone-sensitive *Arabidopsis* mutant named *rcd3* (radical-induced cell death3) was isolated, showing constitutively higher stomatal conductance and deficit in the well-known activations of S-type guard cell anion channels by Ca^2+^ or abscisic acid (ABA) [31,32,35,36,37,38]. This mutant was renamed *SLAC1* (slow anion channel-associated 1), and afterwards, *SLAC1* gene was shown to encode a guard cell expressed S-type anion channel [38,39].

SLAC1 is predicted to be a membrane protein with ten transmembrane helices which, based on the structure of its bacterial homologue, forms a symmetrical trimer [38,40,41]. Usually in its inactive state, in which a phenylalanine residue at position 450 (Phe450) blocks its channel pore, SLAC1 is only activated when it is phosphorylated by certain kinases and a conformational change allows the removal of the Phe450 residue [40,42,43,44]. Various kinases are involved in this activation of SLAC1, including SnRK (sucrose non-fermenting-related kinase), LRR-RLK (leucine-rich repeat kinase), MPK (mitogen-activated protein kinase), CPK/CDPK (calcium dependent kinase), CBL (calcineurin-B like protein) and CIPK (CBL-interacting protein kinase) (Table 1; Figure 1) [45,46,47,48,49,50,51,52,53,54,55].

Drought/salt responsive stomatal closure occurs by drought-driven synthesis of ABA, which is an essential signaling hormone for some activators of SLAC1. SnRK2.6 also known as OST1 (Open stomata 1), identified through functional screening of *Arabidopsis* mutants, was the first reported kinase to activate SLAC1 [56,57]. Activity of OST1 is usually suppressed by the protein phosphatase ABI1. Application of ABA induces protein complex formation of ABI1 and ABA receptor proteins PYR/PYL/RCAR, which subsequently inactivates ABI1 and allows OST1 to phosphorylate Ser120 residue of SLAC1 [45,46,56,57,58,59,60,61]. OST1 also possesses the ability to phosphorylate and inactivate K^+^ in channel KAT1, further emphasizing its significance in the closing of stomata [62,63]. In addition to this, a member of LRR-RLK named GHR1 (guard cell hydrogen peroxide-resistant 1) was identified as an alternative key kinase in ABA-dependent regulation of SLAC1 activity. In contrast to OST1, GHR1 interacts with protein phosphatase ABI2 and not ABI1, suggesting that GHR1 acts in parallel with OST1 upon ABA-induced stomatal closure [64]. OST1, activated by ABA, is also capable of promoting ROS (reactive oxygen species) production in guard cell by phosphorylating NADPH oxidase RbohD and RbohF [63,65,66]. Two MPKs, MPK9 and MPK12, are known to mediate ROS-induced S-type channel activation in guard cell [54,67]. Murata et al. reported that ABI2, not ABI1, was inhibited by ROS, suggesting a ROS-mediated indirect enhancement of GHR1 activity [51,58,64]. In a recent study, another kinase BAK1 (Brassinosteroid insensitive 1-associated receptor kinase 1) was shown to directly form a complex with OST1 and stimulate stomatal closure in an ABA-dependent manner. BAK1 functions upstream of ROS production, and its complex formation with OST1 is inhibited by ABI1 [68]. In addition, accumulation of ROS (particularly H_2_O_2_) enhances synthesis of nitric oxide (NO) by NR1 (nitrate reductase 1) [69,70]. NO can provide both positive or negative feedback in stomatal closure [70,71,72,73]. NO induces phosphatidic acid (PA) production via activation of phospholipase C or D, which in turn inhibits ABI1 and activates RbohD/RbohF [74,75,76,77]. On the other hand, accumulation of NO triggers degradation of ABA receptors PYR/PYL/RCAR via tyrosine nitration, and attenuation of OST1 and RbohD activity by *S*-nitrosylation [78,79,80].

ABA also stimulates increases in cytosolic Ca^2+^ concentration. This is achieved by the release of Ca^2+^ from intracellular stores and Ca^2+^ influx into the cell via inward Ca^2+^ transporters [31,113,114,115,116,117,118]. One of the known mechanisms is the activation of hyperpolarization-activated Ca^2+^-permeable I_Ca_ channels by ROS (which is in turn increased by ABA, as described above) [58,119,120]. The elevated Ca^2+^ concentration activates CBL1 or CBL9, both of which form a complex with CIPK26 and phosphorylate RbohF, forming a positive feedback loop in ROS production [48,121]. Another group of Ca^2+^-activated kinases, CPK4, 5, 6 and 11, regulates RbohD in a similar manner [122,123]. Accumulation of ROS promotes synthesis of NO, which is capable of releasing Ca^2+^ from intracellular Ca^2+^ stores [72,124,125]. These mechanisms allow the increase of cytosolic Ca^2+^ concentration and the activation of various Ca^2+^ dependent kinases in guard cell, which subsequently phosphorylate and activate SLAC1. To date, the following kinases are known to participate in activation of SLAC1: CPK3, 6, 21, 23, CIPK23 (with CBL1 or CBL9) and CIPK11 (with CBL5) [55,61,82,83,126]. All these kinases are confirmed to activate SLAC1 in *Xenopus* oocyte, and they recognize different phosphorylation sites from OST1 (for instance, CPK6 phosphorylates Ser59 of SLAC1) [61,83]. However these kinases, like OST1, are confirmed to be inhibited by ABI1 and ABI2, suggesting a complex crosstalk between SLAC1-activating kinases [61,82,83,127]. The four CPKs seem to display differences in Ca^2+^ affinity, since CPK6 and CPK23 enable SLAC1 to emit anion currents at resting Ca^2+^ concentration in oocytes, while CPK3 and CPK21 requires deletion of their EF-hands (which renders them constitutively active) [82,126]. Patch clamp analysis revealed the severe reduction of S-type anion channel current amplitude in guard cell of *cpk23* [82]. In addition, the ABA-induced activation of guard cell S-type anion channels was disrupted in *cpk3cpk6* double or *cpk5cpk6cpk11cpk23* quadruple mutant [128,129]. While *cpk3cpk6* and *cpk5cpk6cpk11cpk23* mutant plants showed impaired ABA-induced stomatal closure, stomatal behavior in *cipk23* and *cbl1cbl9* was somehow the opposite, displaying reduced leaf water transpiration and enhanced drought tolerance [55,109,128,129]. The effect of *cpk23* mutation is still under debate since conflicting results are reported [130,131]. *cpk21* mutant showed enhanced tolerance in osmotic stress and no change in stomatal conductance, even though CPK21 was confirmed to activate GORK, an outward guard cell K^+^ channel that works synergistically with SLAC1 upon stomatal closure [111,131,132]. These somewhat confusing results can be explained by functional overlapping and compensation of CPKs. For example, gene expression of *CPK23* is upregulated in *cpk21* mutant plant [132]. As for CIPK23, this kinase is known to activate various channels other than SLAC1, including inward K^+^ channel AKT1 [108,109,110]. It was therefore speculated, that CIPK23 activates other channels like AKT1 rather than SLAC1 in vivo, resulting in negative regulation of ABA signaling in guard cells [3,45]. Recently, *N*-myristoylation and *S*-acylation at the N-terminus of CPK or CBL, was identified as an essential modification of CPK6 and CBL5-CIPK11 upon their activation of SLAC1, in terms of their PM recruitment [55]. Conservation of *N*-myristoylation and *S*-acylation motif among CPKs and CBLs suggest this mechanism as a common requirement for ion channel regulation by these kinases at the membrane [133,134,135,136,137].

In addition to ABA, methyl jasmonate (MeJA) and salicylic acid (SA) also induces stomatal closure via the regulation of S-type anion channel (Figure 1). MeJA-induced stomatal closure is dependent on the activation of guard cell PM H^+^-ATPase, a process that is mediated by F-box protein COI1 (Coronatine-insensitive 1) [138]. This promotes hyperpolarization of PM and activation of PM Ca^2+^ channels, resulting in an increase of cytosolic Ca^2+^. The elevated Ca^2+^ level results in activation of CPK6, though not of CPK3, 4 or 11, and the consequent SLAC1 activation [139]. MPK9, MPK12 and GHR1 also participate in SLAC1 activation in this pathway, possibly as a consequence of RbohD/RbohF-mediated ROS production by elevated Ca^2+^ [49,64]. Similarly, SA-induced stomatal closure involves ROS production, followed by SLAC1 activation via CPK3, 6, MPK9, MPK12, and GHR1 [54,64,140]. However, this pathway is unique in the way that ROS is produced: it features extracellular ROS produced by SHAM (salicylhydroxamic acid)-sensitive peroxidase, which is then diffused into guard cells [141].

Because stomata work as a major gateway for CO_2_ influx, elevated CO_2_ levels also promote stomatal closure (Figure 1) [39,45]. In this regulation pathway, CO_2_ imported from stomatal pore is first converted into HCO_3_^−^ by two kinds of carbonic anhydrases (CAs), βCA1 and βCA4. Exposure to high CO_2_ condition elevates the intracellular HCO_3_^−^ concentration, stimulating an HCO_3_^−^ sensing component named RHC1 (Resistant to high carbon dioxide 1). This induces the formation of a complex between CA and RHC1, which enables the interaction with, and the subsequent inactivation of HT1 kinase (High leaf temperature 1e), an inhibitor of OST1 and GHR1 [45,81,142,143,144,145,146]. This mechanism would allow SLAC1 activation and the consequent stomatal closure under high CO_2_ condition. In a recent study, CO_2_-permeable aquaporin PIP2;1 was identified as an upstream regulator of βCA1 and βCA4, and MPK4 and MPK12 as additional intermediates for HT1 regulation by HCO_3_^−^ [81,147].

Immediate stomatal closure is necessary during bacterial invasion, for stomata might serve as an entrance for bacteria [148,149]. Melotto et al. revealed that plant closes stomata when exposed to *P. syringae*, *E. coli* and PAMPs (pathogen-associated molecular pattern) [150]. This mechanism was predicted to involve a signal transduction from a receptor like kinase FLS2 (which recognizes flg22, a 22-amino-acid residue stretch of the flagellin protein from *P. syringae*) to some kinases that activate SLAC1 [148,151,152,153]. In recent years it was indicated that OST1, not CPKs, was responsible for this pathogen-induced stomatal closure mediated by SLAC1 [154]. However, it was also confirmed that NADPH oxidases and ABI1 were unnecessary for this pathway, leaving a missing link between FLS2 and OST1 [154]. Several studies report the formation of a complex between FLS2 and BAK1, a possible activator of OST1, so it can be hypothesized that BAK1 mediates the bacterial resistance signal from FLS2 to OST1 [68,84,85]. On the other hand, several researches proposed alternative pathways. Recent research reported that another FLS2-associated kinase, BIK1, directly phosphorylates RbohD, suggesting an OST1-independent ROS production pathway [86,87]. This seem to hold the truth for stomatal closure by another pathogen responsive component, danger-associated peptides (DAMPs), which was confirmed not to require OST1 [155]. Montillet et al. showed another OST1-independent pathway which involves the oxylipin production induced by lipoxygenase 1 (LOX1), resulting in an increased amount of SA (which might be consistent with the reports that MPK9 and MPK12 function in biotic stress response) [54,156,157]. Su et al. proposed a pathway insensitive to coronatine, a compound produced by pathogens, which promotes ‘reopening’ of stomata [153,158]. This model considers the activation of MPK3 and MPK6 and the induced malate/citrate metabolism as key features. However, direct connection to stomatal closure is still under discussion [158].

## 3. SLAHs (SLAC1 Homologues) and Nitrate Transporters

In the *Arabidopsis* genome, four *SLAC1* homologue genes (*SLAH1-4*) are present [38,39]. When focusing on stomatal movements, SLAH3 protein is particularly important among these four, since it is the only SLAH3 confirmed to be expressed in guard cell [88,107]. Unlike SLAC1, which is permeable to both chloride and nitrate, permeability of SLAH3 is strongly restricted to nitrate [88]. Guard cell protoplasts from *slac1-3* mutant plants elicited S-type anion channel current in nitrate-based buffer, but not from the *slah3-1* mutant, suggesting SLAH3, but not SLAC1, is the key component of nitrate-mediated stomatal regulation [88]. This indicates the requirement of both SLAC1 and SLAH3 for full stomatal function, as implied in some studies [154,155]. Most of the Ca^2+^ related SLAC1-activating kinases, including CPK3, CPK6, CPK21, CPK23, and CBL1/CBL9-CIPK23, can also activate SLAH3 [83,88,89]. However, (and significantly), SLAH3 was insensitive to OST1, a major activator of SLAC1 [88]. Surprisingly, it was reported that SLAH3, as well as SLAC1, physically interacts with and inhibits an inward K^+^ channel KAT1 [107]. Taken together, stomatal closure pathway by SLAH3, even though similar to that of SLAC1, harbors some unique features (Figure 1).

Cubero-Font et al. discovered that *SLAH1* gene encodes a channel subunit which forms a heteromer with SLAH3, and renders it permeable to nitrate and chloride [159]. Although no evidence of SLAH1 expression in guard cell have arisen so far, it is interesting to note that SLAH1 expression driven by *SLAC1* promoter can complement the stomatal phenotypes of *slac1* mutation [39,159]. SLAH2, on the other hand, is expressed mainly in roots, and shows S-type channel activity with strict nitrate selectivity [44]. CBL1-CIPK23 and several CPKs, including CPK3 and 21, were identified as the activator of this nitrate channel [44,160]. To this date, data on the function and expression of SLAH4 awaits discovery.

In 2003, Guo et al. demonstrated that CHL1 (also referred to as AtNRT1.1 or AtNPF6.3), a dual-affinity nitrate transporter, is expressed in guard cell and functions in stomatal opening and guard cell nitrate accumulation [161]. CBL1/CBL9-CIPK23 phosphorylates CHL1 and converts its nitrate transport from low-affinity mode to high-affinity mode (which, in turn, results in reduced nitrate intake under sufficient nitrate concentration) [90,91,92,93]. The behavior of CHL1 in stomata has not been reported extensively. The fact that they are both regulated by CBL1/9-CIPK23 suggests that there might be a crosstalk between CHL1 and SLAH (Figure 1).

## 4. Malate Transporters

Even after SLAC1 was identified as a guard cell S-type anion channel, the origin of R-type anion channel currents had remained unknown [38,39]. Then in 2010, several researches identified AtALMT12, a member of aluminum-activated malate transporter family, as a major component of the guard cell R-type anion channels [46,162,163]. AtALMT12 represents an anion channel permeable to malate, chloride and nitrate, and unlike SLAC1 it does not require any kinases for activation [162,163]. However, coexpression of OST1 in *Xenopus* oocytes resulted in further enhancement of AtALMT12 current [97]. AtALMT12 is different from its homologue AtALMT1 in that its activity was not stimulated by Al^3+^ [163,164]. *atalmt12* mutant plants showed partially impaired stomatal closure in response to various stimuli such as CO_2_, ABA and Ca^2+^ [162,163]. AtALMT12 shows rapid inactivation at hyperpolarized membrane potential, in which its cytosolic C-terminal domain serves as a voltage sensor [165].

Three other members of ALMT family reside in *Arabidopsis* guard cell tonoplast; AtALMT4, 6 and 9 (Figure 1) [95,96,166,167,168]. AtALMT6 was identified as a Ca^2+^-activated channel permeable to malate and fumarate, and its activity modulated by vacuolar pH [95]. This channel can mediate malate influx upon tonoplast hyperpolarization, and efflux upon depolarization [95]. *Atalmt6* mutant plant displayed reduced malate current in guard cell vacuole. However, no obvious phenotypic difference was observed compared to WT plants [95]. AtALMT9, on the other hand, acts as a chloride efflux channel activated by cytosolic malate, and the phenotype of the mutant plant evidenced its role in stomatal opening [96]. Recently, AtALMT4 was shown to be another tonoplast ALMT, that can mediate anion influx and efflux, regulated by phosphorylation at Ser382 residue [94]. Though its function shows some similarity to AtALMT6, *Atalmt4* mutant plant shows obvious impairment in ABA-induced stomatal closure [94].

The significance of malate during stomatal movements has been implied decades before the identification of ALMTs described above [47,169,170,171,172]. This includes malate release from mesophyll cells and malate synthesis in guard cells [22,47,171,172]. In 2008, an ABC transporter AtABCB14, was reported to mediate malate influx across guard cell PM, and confirmed to participate in stomatal opening [98]. Together, these data suggest an elaborate regulation of stomatal movement by malate.

## 5. Other Anion Channels Involved in Stomatal Closure

In addition to the channels described above, several anion channels from other families have been shown to compete in stress-induced stomatal movement. CLC is a family of chloride channels that can be found ubiquitously among bacteria, animals, and plants [46,173,174,175,176]. Seven members of *CLC* gene, *AtCLCa-g*, are present in *Arabidopsis* genome [177,178]. Among these seven, AtCLCc is particularly expressed in guard cell tonoplast. Though its detailed function remains unveiled, the phenotype of *clcc* mutants postulate its participation in stomatal response to ABA and NaCl through modulation of chloride/nitrate homeostasis [102]. Wege et al. stated that another CLC member AtCLCa, originally identified as a tonoplast nitrate/proton antiporter, was required for ABA-induced stomatal closure or inhibition of stomatal opening [99,100]. AtCLCa mediates nitrate influx at hyperpolarization and efflux upon depolarization [99,100]. Although ATP binding to the C-terminus of AtCLCa inhibits its transport activity, its activity can be resumed by phosphorylation via OST1, suggesting some crosstalk between SLAC1 and AtALMT12 regulation [100,101,168]. Two members from *Arabidopsis* detoxification efflux carrier (DTX)/Multidrug and toxic compound extrusion (MATE) family, DTX33 and 35, were recently identified as an additional anion channel residing in guard cell tonoplast [103,104,105]. Both channels exhibit vacuolar chloride influx in various types of cells, including guard cell, and mutation of their genes results in impaired stomatal opening [105]. Andrés et al. reported that the dynamic structure change of guard cell vacuole itself is necessary for proper regulation of stomata movement, and that this is mediated by tonoplast K^+^/H^+^ antiporter NHX1 and NHX2, and, possibly, by the yet unidentified Na^+^/K^+^ antiporter [179]. It was later proposed that CBL2, CBL3, CIPK9 and CIPK17 might contribute in this NHX1/2-mediated convolution of vacuole [112].

Studies from the 1980s demonstrated that SO_2_ can also promote stomatal closure, which was most recently concluded as being a result of non-apoptotic cell death caused by the accumulation of H_2_SO_3_ [180,181,182]. Another research implicates sulfate as an important element for drought-induced stomatal closure [106]. Sulfate, uptaken from roots through sulfate transporter SULTRs, had long been known as an essential macronutrient for plant growth and development [27,28,183]. SULTR family is divided into five groups [184]. Recently the essential role of sulfate in ABA synthesis was elucidated, suggesting the participation of SULTRs in abiotic stress response [29,30,185,186]. Malcheska et al. reported stomatal closure induced by apoplastic sulfate, and activation of AtALMT12 by sulfate application [106]. This study also proposed that sulfate induces enhanced ABA synthesis inside the guard cell, possibly involving chloroplastic SULTR3;1 as a key transporter (Figure 1) [106].

## 6. Conclusions

Stomatal movement triggered by biotic and abiotic stress is regulated by an intricate mechanism involving various guard cell expressed ion channels and their regulator components (kinases, phytohormone receptor, etc.). The circulation and accumulation of anions in plants require a large number of transport systems. The extensive identification on anion channels and transporters described in this mini-review has filled in many of the gaps which had precluded the full understanding of the mechanism of ion homeostasis and cellular adaptation against harsh salinity stress. Further progress is a necessary demand for the successful modification of plant stress tolerance to global environmental changes.

## Figures and Tables

**Figure 1 plants-08-00009-f001:**
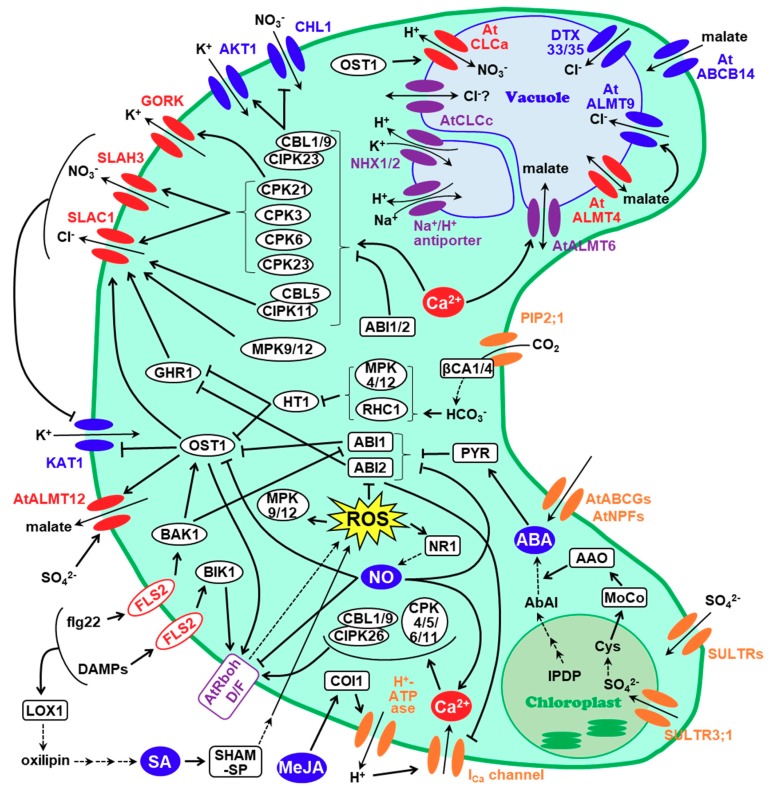
Schematic model of ion channel regulation during stress induced stomatal movement in *Arabidopsis thaliana*. Thin arrow, influx/efflux of compounds; thick arrow, activation or inhibition; broken arrow, breakdown or biosynthesis. Channels/Transporters, which have been evidenced to control stomatal opening or closure, are shown in red and blue, respectively. Channels shown in purple are the ones that contribute to both opening and closure, or their role in stomatal regulation still remains ambiguous. Other channels/transporters are shown in orange. Abbreviations; SHAM-SP, SHAM-sensitive peroxidase; IPDP, isopentenyl diphosphate; AbAl, abscisic aldehyde; MoCo, molybdenum cofactor; AAO, abscisic aldehyde oxidase.

**Table 1 plants-08-00009-t001:** List of guard cell expressed ion channels/transporters and their regulator components.

Name	Subcellular Localization	Function	Activation or Deactivation	Regulatory Components	Reference
SLAC1	PM	Cl^−^ efflux	A	OST1	[56,57,61]
			A	GHR1	[64,81]
			A	MPK9/12	[54,67]
			A	CPK3/6/21/23	[61,82,83]
			A	CBL1/9-CIPK23	[83]
			A	CBL5-CIPK11	[55]
			A	BAK1 (via OST1 phosphorylation)	[68,84,85]
			A	BIK1 (via ROS production)	[86,87]
SLAH3	PM	NO_3_^−^ efflux	A	CPK3/6/21/23	[83,88,89]
			A	CBL1/9-CIPK23	[83]
CHL1	PM	NO_3_^−^ influx	D	CBL1/9-CIPK23 (via conversion of nitrate transport mode)	[90,91,92,93]
AtALMT4	Tonoplast	Malate efflux/influx			[94]
AtALMT6			A	Ca^2+^	[95]
AtALMT9	Tonoplast	Cl^−^ influx	A	Cytosolic malate	[96]
AtALMT12	PM	malate efflux	A	OST1	[97]
AtABCB14	PM	malate influx			[98]
AtCLCa	Tonoplast	H^+^ efflux/NO_3_^−^ influx	A	CBL1/9-CIPK23	[99,100,101]
AtCLCc	Tonoplast				[102]
DTX33	Tonoplast	Cl^−^ influx			[103,104,105]
DTX35					
SULTR3;1	Chloroplast	SO_4_^−^ influx			[106]
KAT1	PM	K^+^ influx	D	OST1	[62,63]
			D	SLAC1, SLAH3	[107]
AKT1	PM	K^+^ influx	D	CBL1/9-CIPK23	[108,109,110]
GORK	PM	K^+^ efflux	A	CPK21	[111]
NHX1	Tonoplast	H^+^ efflux/K^+^ influx	A	CBL2/3, CIPK9/17	[112]
NHX2					

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
