# Peer review of "Guard Cell Membrane Anion Transport Systems and Their Regulatory Components: An Elaborate Mechanism Controlling Stress-Induced Stomatal Closure"

_plants, 2019, doi:10.3390/plants8010009_

Round 1

Reviewer 1 Report

The authors review current literature on stomata guard cell ion transporters and their regulators. 

This is a well-written mini-review, and I have only minor suggestions regarding English language:

L38 "in the shoot" instead of "in shoot"

L39: "causes a decreased" instead of "cause decreased"

L275: please check the word "mechanism", it looks unclear in my pdf version, and please check spelling of this word

L277: :anions in plants" instead of "anion in plant"

L279: "precluded" instead of "preclude"

Reviewer 2 Report

The current review paper summarised the recent progress in anion channels and their response to abiotic stress and stomatal regulation. The unique angle of this paper is on the protein kinase regulation of anion channels, which has not be comprehensive reviewed. Here are a few suggestions:

) The scope of title is narrow for a review paper. Also, the authors discussed carriers and transporters in the manuscript. A better title should be used.

) A summary Table should be included with references to the key literature about the anion transporters and interacting proteins (e.g. CDPKs).

) In Figure 1, a key secondary messenger nitric oxide (NO) is missing. NO also regulates ion channels for stomatal closure, which can not be ignored.

Round 2

Reviewer 1 Report

authors have followed suggested minor revisions